# Causal Estimation for Text Data with (Apparent) Overlap Violations

Lin Gui[1] and Victor Veitch[1,2]

[1]*The University of Chicago*
[2]*Google Research*

## Abstract

Consider the problem of estimating the causal effect of some attribute of a text document; for example: what effect does writing a polite vs. rude email have on response time? To estimate a causal effect from observational data, we need to adjust for confounding aspects of the text that affect both the treatment and outcome—e.g., the topic or writing level of the text. These confounding aspects are unknown a priori, so it seems natural to adjust for the entirety of the text (e.g., using a transformer). However, causal identification and estimation procedures rely on the assumption of overlap: for all levels of the adjustment variables, there is randomness leftover so that every unit could have (not) received treatment. Since the treatment here is itself an attribute of the text, it is perfectly determined, and overlap is apparently violated. The purpose of this paper is to show how to handle causal identification and obtain robust causal estimation in the presence of apparent overlap violations. In brief, the idea is to use supervised representation learning to produce a data representation that preserves confounding information while eliminating information that is only predictive of the treatment. This representation then suffices for adjustment and satisfies overlap. Adapting results on non-parametric estimation, we find that this procedure is robust to conditional outcome misestimation, yielding a low-absolute-bias estimator with valid uncertainty quantification under weak conditions. Empirical results show strong improvements in bias and uncertainty quantification relative to the natural baseline. Code, demo data and a tutorial are available at https://github.com/gl-ybnbxb/TI-estimator.

## 1 Introduction

We consider the problem of estimating the causal effect of an attribute of a passage of text on some downstream outcome. For example, what is the effect of writing a polite or rude email on the amount of time it takes to get a response? In principle, we might hope to answer such questions with a randomized experiment. However, this can be difficult in practice—e.g., if poor outcomes are costly or take long to gather. Accordingly, in this paper, we will be interested in estimating such effects using observational data.

There are three steps to estimating causal effects using observational data (See Chapter 36 Murphy (2023)). First, we need to specify a concrete causal quantity as our estimand. That is, give a formal quantity target of estimation corresponding to the high-level question of interest. The next step is causal identification: we need to prove that this causal estimator can, in principle, be estimated using only observational data. The standard approach for identification relies on adjusting for confounding variables that affect both the treatment and the outcome. For identification to hold, our adjustment variables must satisfy two conditions: unconfoundedness and overlap. The former requires the adjustment variables contain sufficient information on all common causes. The latter requires that the adjustment variable does not contain enough information about treatment assignment to let us perfectly predict it. Intuitively, to disentangle the effect of treatment from the effect of confounding, we must observe each treatment state at all levels of confounding. The final

step is estimation using a finite data sample. Here, overlap also turns out to be critically important as a major determinant of the best possible accuracy (asymptotic variance) of the estimator Chernozhukov et al. (2016).

Since the treatment is a linguistic property, it is often reasonable to assume that text data has information about all common causes of the treatment and the outcome. Thus, we may aim to satisfy unconfoundedness in the text setting by adjusting for all the text as the confounding part. However, doing so brings about overlap violation. Since the treatment is a linguistic property determined by the text, the probability of treatment given any text is either 0 or 1. The polite/rude tone is determined by the text itself. Therefore, overlap does not hold if we naively adjust for all the text as the confounding part. This problem is the main subject of this paper. Or, more precisely, our goal is to find a causal estimand, causal identification conditions, and a robust estimation procedure that will allow us to effectively estimate causal effects even in the presence of such (apparent) overlap violations.

In fact, there is an obvious first approach: simply use a standard plug-in estimation procedure that relies only on modeling the outcome from the text and treatment variables. In particular, do not make any explicit use of the propensity score, the probability each unit is treated. Pryzant et al. (2020) use an approach of this kind and show it is reasonable in some situations. Indeed, we will see in Sections 3 and 4 that this procedure can be interpreted as a point estimator of a controlled causal effect. Even once we understand what the implied causal estimand is, this approach has a major drawback: the estimator is only accurate when the text-outcome model converges at a very fast rate. This is particularly an issue in the text setting, where we would like to use large, flexible, deep learning models for this relationship. In practice, we find that this procedure works poorly: the estimator has significant absolute bias and (the natural approach to) uncertainty quantification almost never includes the estimand true value; see Section 5.

The contribution of this paper is a method for robustly estimating causal effects in text. The main idea is to break estimation into a two-stage procedure, where in the first stage we learn a representation of the text that preserves enough information to account for confounding, but throws away enough information to avoid overlap issues. Then, we use this representation as the adjustment variables in a standard double machine-learning estimation procedure Chernozhukov et al. (2016; 2017a). To establish this method, the contributions of this paper are:

1. We give a formal causal estimand corresponding to the text-attribute question. We show this estimand is causally identified under weak conditions, even in the presence of apparent overlap issues.

2. We show how to efficiently estimate this quantity using the adapted double-ML technique just described. We show that this estimator admits a central limit theorem at a fast ($\sqrt{n}$) rate under weak conditions on the rate at which the ML model learns the text-outcome relationship (namely, convergence at $n^{1/4}$ rate). This implies absolute bias decreases rapidly, and an (asymptotically) valid procedure for uncertainty quantification.

3. We test the performance of this procedure empirically, finding significant improvements in bias and uncertainty quantification relative to the outcome-model-only baseline.

**Related work** The most related literature is on causal inference with text variables. Papers include treating text as treatment Pryzant et al. (2020); Wood-Doughty et al. (2018); Egami et al. (2018); Fong & Grimmer (2016); Wang & Culotta (2019); Tan et al. (2014)), as outcome Egami et al. (2018); Sridhar & Getoor (2019), as confounder Veitch et al. (2019); Roberts et al. (2020); Mozer et al. (2020); Keith et al. (2020), and discovering or predicting causality from text del Prado Martin & Brendel (2016); Tabari et al. (2018); Balashankar et al. (2019); Mani & Cooper (2000). There are also numerous applications using text to adjust for confounding (e.g., Olteanu et al., 2017; Hall, 2017; Kiciman et al., 2018; Sridhar et al., 2018; Sridhar & Getoor, 2019; Saha et al., 2019; Karell & Freedman, 2019; Zhang et al., 2020). Of these, Pryzant et al. (2020) also address non-parametric es-

timation of the causal effect of text attributes. Their focus is primarily on mismeasurement of the treatments, while our motivation is robust estimation.

This paper also relates to work on causal estimation with (near) overlap violations. D'Amour et al. (2021) points out high-dimensional adjustment (e.g., Rassen et al., 2011; Louizos et al., 2017; Li et al., 2016; Athey et al., 2017) suffers from overlap issues. Extra assumptions such as sparsity are often needed to meet the overlap condition. These results do not directly apply here because we assume there exists a low-dimensional summary that suffices to handle confounding.

D'Amour & Franks (2021) studies summary statistics that suffice for identification, which they call deconfounding scores. The supervised representation learning approach in this paper can be viewed as an extremal case of the deconfounding score. However, they consider the case where ordinary overlap holds with all observed features, with the aim of using both the outcome model and propensity score to find efficient statistical estimation procedures (in a linear-gaussian setting). This does not make sense in the setting we consider. Additionally, our main statistical result (robustness to outcome model estimation) is new.

## 2 NOTATION AND PROBLEM SETUP

We follow the causal setup of Pryzant et al. (2020). We are interested in estimating the causal effect of treatment $A$ on outcome $Y$. For example, how does writing a negative sentiment ($A$) review ($X$) affect product sales ($Y$)? There are two immediate challenges to estimating such effects with observed text data. First, we do not actually observe $A$, which is the intent of the writer. Instead, we only observe $\tilde{A}$, a version of $A$ that is inferred from the text itself. In this paper, we will assume that $A = \tilde{A}$ almost surely—e.g., a reader can always tell if a review was meant to be negative or positive. This assumption is often reasonable, and follows Pryzant et al. (2020). The next challenge is that the treatment may be correlated with other aspects of the text ($Z$) that are also relevant to the outcome— e.g., the product category of the item being reviewed. Such $Z$ can act as confounding variables, and must somehow be adjusted for in a causal estimation problem.

Each unit $(A_i, Z_i, X_i, Y_i)$ is drawn independently and identically from an unknown distribution $P$. Figure 1 shows the causal relationships among variables, where solid arrows represent causal relations, and the dotted line represents possible correlations between two variables. We assume that text $X$ contains all common causes of $\tilde{A}$ and the outcome $Y$.

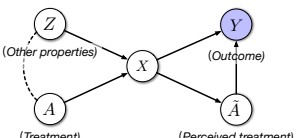

**Figure 1:** The causal DAG of the problem. A writer writes a text document $X$ based on linguistic properties $A$ and $Z$, where $A$ is the treatment in the causal problem. $A$ and $Z$ cannot be observed directly in data and can only be seen via text. The dotted line represents possible correlation between $A$ and $Z$. A reader perceives the treatment $\tilde{A}$ from the text. The perceived treatment $\tilde{A}$ together with contents of $X$ determine the outcome $Y$.

## 3 IDENTIFICATION AND CAUSAL ESTIMAND

The first task is to translate the qualitative causal question of interest—what is the effect of $A$ on $Y$—into a causal estimand. This estimand must both be faithful to the qualitative question and be identifiable from observational data under reasonable assumptions. The key challenges here are that we only observe $\tilde{A}$ (not $A$ itself), there are unknown confounding variables influencing the text, and $\tilde{A}$ is a deterministic function of the text, leading to overlap violations if we naively adjust for all the text. Our high-level idea is to split the text into abstract (unknown) parts depending on whether they are confounding—affect both $\tilde{A}$ and $Y$—or whether they affect $\tilde{A}$ alone. The part of the text that affects only $\tilde{A}$ is not necessary for causal adjustment, and can be thrown away. If this part contains "enough" information about $\tilde{A}$, then throwing it away can eliminate our ability to perfectly predict $\tilde{A}$, thus fixing the overlap issue. We now turn to formalizing this idea, showing how it can be used to define an estimand and to identify this estimand from observational data.

**Causal model**  The first idea is to decompose the text into three parts: one part affected by only $A$, one part affected interactively by $A$ and $Z$, and another part affected only by $Z$. We use $X_A$, $X_{A \wedge Z}$ and $X_Z$ to denote them, respectively; see Figure 2 for the corresponding causal model. Note that there could be additional information in the text in addition to these three parts. However, since they are irrelevant to both $A$ and $Z$, we do not need to consider them in the model.

**Controlled direct effect (CDE)**  The treatment $A$ affects the outcome through two paths. Both "directly" through $X_A$—the part of the text determined just by the treatment—and also through a path going through $X_{A \wedge Z}$—the part of the text that relies on interaction effects with other factors. Our formal causal effect aims at capturing the effect of $A$ through only the first, direct, path.

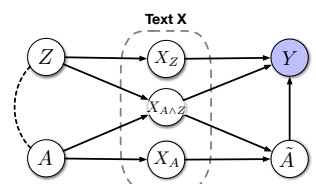

**Text X**

**Figure 2:** A more sophisticated causal model with the decomposition of text $X$. $X_A$, $X_{A \wedge Z}$, and $X_Z$ are parts of the text affected by only $A$, both $A$ and $Z$, and only $Z$, respectively. $A$ and $Z$ are linguistic properties a writer based on and thus cannot be observed directly from data. When investigating the causal relationship between $\tilde{A}$ and $Y$, $(X_{A \wedge Z}, X_Z)$ is a confounding part satisfying both unconfoundedness and overlap.

$$\mathrm{CDE} := \mathbb{E}_{X_{A \wedge Z}, X_Z | A = 1} \Big[ \mathbb{E}[Y \mid X_{A \wedge Z}, X_Z, \mathrm{do}(A = 1)]$$
$$- \mathbb{E}[Y \mid X_{A \wedge Z}, X_Z, \mathrm{do}(A = 0)] \Big]. \quad (3.1)$$

Here, do is Pearl's do notation, and the estimand is a variant of the controlled direct effect (Pearl, 2009). Intuitively, it can be interpreted as the expected change in the outcome induced by changing the treatment from 1 to 0 while keeping part of the text affected by $Z$ the same as it would have been had we set $A = 1$. This is a reasonable formalization of the qualitative "effect of $A$ on $Y$". Of course, it is not the only possible formalization. Its advantage is that, as we will see, it can be identified and estimated under reasonable conditions.

**Identification**  To identify CDE we must rewrite the expression in terms of observable quantities. There are three challenges: we need to get rid of the do operator, we don't observe $A$ (only $\tilde{A}$), and the variables $X_{A \wedge Z}, X_Z$ are unknown (they are latent parts of $X$).

Informally, the identification argument is as follows. First, $X_{A \wedge Z}, X_Z$ block all backdoor paths (common causes) in Figure 2. Moreover, because we have thrown away $X_A$, we now satisfy overlap. Accordingly, the do operator can be replaced by conditioning following the usual causal-adjustment argument. Next, $A = \tilde{A}$ almost surely, so we can just replace $A$ with $\tilde{A}$. Now, our estimand has been reduced to:

$$\widetilde{\mathrm{CDE}} := \mathbb{E}_{X_{A \wedge Z}, X_Z | \tilde{A} = 1} \big[ \mathbb{E}[Y \mid X_{A \wedge Z}, X_Z, \tilde{A} = 1] - \mathbb{E}[Y \mid X_{A \wedge Z}, X_Z, \tilde{A} = 0)] \big]. \quad (3.2)$$

The final step is to deal with the unknown $X_{A \wedge Z}, X_Z$. To fix this issue, we first define the *conditional outcome Q* according to:

$$Q(\tilde{A}, X) := \mathbb{E}(Y \mid \tilde{A}, X). \quad (3.3)$$

A key insight here is that, subject to the causal model in Figure 2, we have $Q(\tilde{A}, X) = \mathbb{E}(Y \mid \tilde{A}, X_{A \wedge Z}, X_Z)$. But this is exactly the quantity in (3.2). Moreover, $Q(\tilde{A}, X)$ is an observable data quantity (it depends only on the distribution of the observed quantities). In summary:

**Theorem 1.** *Assume the following:*
*1. (Causal structure) The causal relationships among $A$, $\tilde{A}$, $Z$, $Y$, and $X$ satisfy the causal DAG in Figure 2;*
*2. (Overlap) $0 < \mathrm{P}(A = 1 \mid X_{A \wedge Z}, X_Z) < 1$;*
*3. (Intention equals perception) $A = \tilde{A}$ almost surely with respect to all interventional distributions. Then, the CDE is identified from observational data as*

$$\mathrm{CDE} = \tau^{\mathrm{CDE}} := \mathbb{E}_{X | \tilde{A} = 1} \big[ \mathbb{E}[Y \mid \eta(X), \tilde{A} = 1] - \mathbb{E}[Y \mid \eta(X), \tilde{A} = 0] \big], \quad (3.4)$$

*where $\eta(X) := (Q(0, X), Q(1, X))$.*

The proof is in Appendix B.

We give the result in terms of an abstract sufficient statistic $\eta(X)$ to emphasize that the actual conditional expectation model is not required, only some statistic that is informationally equivalent. We emphasize that, regardless of whether the overlap condition holds or not, the propensity score of $\eta(X)$ is accessible and meaningful. Therefore, we can easily identify when identification fails as long as $\eta(X)$ is well-estimated.

## 4 METHOD

Our ultimate goal is to draw a conclusion about whether the treatment has a causal effect on the outcome. Following the previous section, we have reduced this problem to estimating $\tau^{\text{CDE}}$, defined in Theorem 1. The task now is to develop an estimation procedure, including uncertainty quantification.

### 4.1 OUTCOME ONLY ESTIMATOR

We start by introducing the naive outcome only estimator as a first approach to CDE estimation. The estimator is adapted from Pryzant et al. (2020). The observation here is that, taking $\eta(X) = (Q(0,X), Q(1,X))$ in (3.4), we have

$$\tau^{\text{CDE}} = \mathbb{E}_{X|A=1}\left[\, \mathbb{E}(Y \mid A = 1, X) - \mathbb{E}(Y \mid A = 0, X)\, \right]. \tag{4.1}$$

Since $Q(A, X)$ is a function of the whole text data $X$, it is estimable from observational data. Namely, it is the solution to the square error risk:

$$Q = \arg\min_{\tilde{Q}} \mathbb{E}[(Y - \tilde{Q}(A,X))^2]. \tag{4.2}$$

With a finite sample, we can estimate $Q$ as $\hat{Q}$ by fitting a machine-learning model to minimize the (possibly regularized) square error empirical risk. That is, fit a model using mean square error as the objective function. Then, a straightforward estimator is:

$$\hat{\tau}^Q := \frac{1}{n_1} \sum_{i:A_i=1} \hat{Q}_1(X_i) - \hat{Q}_0(X_i), \tag{4.3}$$

where $n_1$ is the number of treated units.

It should be noted that the model for $Q$ is not arbitrary. One significant issue for those models which directly regress $Y$ on $A$ and $X$ is when overlap does not hold, the model could ignore $A$ and only use $X$ as the covariate. As a result, we need to choose a class of models that force the use of the treatment $A$. To address this, we use a two-headed model that regress $Y$ on $X$ for $A = 0/1$ separately in the conditional outcome learning model (See Section 4.2 and Figure 3).

As discussed in the introduction Section 1, this estimator yields a consistent point estimate, but does not offer a simple approach for uncertainty quantification. A natural guess for an estimate of its variance is:

$$\hat{\text{var}}(\hat{\tau}^Q) := \frac{1}{n} \hat{\text{var}}(\hat{Q}_1(X_i) - \hat{Q}_0(X_i) \mid \hat{Q}). \tag{4.4}$$

That is, just compute the variance of the mean conditional on the fitted model. However, this procedure yields asymptotically valid confidence intervals only if the outcome model converges extremely quickly; i.e., if $\mathbb{E}[(\hat{Q} - Q)^2]^{\frac{1}{2}} = o(n^{-\frac{1}{2}})$. We could instead bootstrap, refitting $\hat{Q}$ on each bootstrap sample. However, with modern language models, this can be prohibitively computationally expensive.

### 4.2 TREATMENT IGNORANT EFFECT ESTIMATION (TI-ESTIMATOR)

Following Theorem 1, it suffices to adjust for $\eta(X) = (Q(0,X), Q(1,X))$. Accordingly, we use the following pipeline. We first estimate $\hat{Q}_0(X)$ and $\hat{Q}_1(X)$ (using a neural language model), as with the outcome-only estimator. Then, we take $\hat{\eta}(X) := (\hat{Q}_0(X), \hat{Q}_1(X))$ and estimate $\hat{g}_\eta \approx P(A = 1 \mid \hat{\eta})$. That is, we estimate the propensity score corresponding to the estimated representation. Finally, we plug the estimated $\hat{Q}$ and $\hat{g}_\eta$ into a standard double machine learning estimator (Chernozhukov et al., 2016).

We describe the three steps in detail.

**Q-Net** In the first stage, we estimate the conditional outcomes and hence obtain the estimated two-dimensional confounding vector $\hat{\eta}(X)$. For concreteness, we will use the dragonnet architecture of Shi et al. (2019). Specifically, we train DistilBERT (Sanh et al., 2019) modified to include three heads, as shown in Figure 3. Two of the heads correspond to $\hat{Q}_0(X)$ and $\hat{Q}_1(X)$ respectively. As discussed in the Section 4.1, applying two heads can force the model to use the treatment $A$. The final head is a single linear layer predicting the treatment. This propensity score prediction head can help prevent (implicit) regularization of the model from throwing away $X_{A\wedge Z}$ information that is necessary for identification. The output of this head is not used for the estimation since its purpose is to force the DistilBERT representation to preserve all confounding information. This has been shown to improve causal estimation (Shi et al., 2019; Veitch et al., 2019).

We train the model by minimizing the objective function

$$\mathcal{L}(\theta; \mathbf{X}) = \frac{1}{n}\sum_i \left[\left(\hat{Q}_{a_i}(x_i; \theta) - y_i\right)^2 + \alpha\text{CrossEntropy}\left(a_i, g_u(x_i)\right) + \beta\mathcal{L}_{\text{mlm}}(x_i)\right], \quad (4.5)$$

where $\theta$ are the model parameters, $\alpha$, $\beta$ are hyperparameters and $\mathcal{L}_{\text{mlm}}(\cdot)$ is the masked language modeling objective of DistilBERT.

There is a final nuance. In practice, we split the data into $K$-folds. For each fold $j$, we train a model $\hat{Q}_{-j}$ on the other $K-1$ folds. Then, we make predictions for the data points in fold $j$ using $\hat{Q}_{-j}$. Slightly abusing notation, we use $\hat{Q}_a(x)$ to denote the predictions obtained in this manner.

**Propensity score estimation** Next, we define $\hat{\eta}(x) := (\hat{Q}_0(x), \hat{Q}_1(x))$ and estimate the propensity score $\hat{g}_\eta(x) \approx P(A = 1 \mid \hat{\eta}(x))$. To do this, we fit a non-parametric estimator to the binary classification task of predicting $A$ from $\hat{\eta}(X)$ in a cross fitting or K-fold fashion. The important insight here is that since $\hat{\eta}(X)$ is

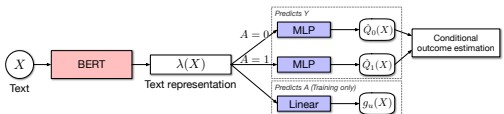

**Figure 3:** The architecture of Q-Net follows the dragonnet (Shi et al., 2019) for estimation of $\hat{Q}$. Specifically, given representations $\lambda(X)$ from input text data, the Q-Net predicts $Y$ for samples with $A = 0$ and $A = 1$ using two separate heads. A third head predicting $A$ is also included for training, though the predictions are not used for estimation. Parameters in DistilBERT and three prediction heads are trained together in an end-to-end manner.

2-dimensional, non-parametric estimation is possible at a fast rate. In Section 5, we try several methods and find that kernel regression usually works well.

We also define $g_\eta(X) := P(A = 1 \mid \eta(X))$ as the idealized propensity score. The idea is that as $\hat{\eta} \to \eta$, we will also have $\hat{g}_\eta \to g_\eta$ so long as we have a valid non-parametric estimate.

**CDE estimation** The final stage is to combine the estimated outcome model and propensity score into a CDE estimator. To that end, we define the influence curve of $\tau^{\text{CDE}}$ as follows:

$$\phi(X; Q, g_\eta, \tau^{\text{CDE}}) := \frac{A \cdot (Y - Q(0, X))}{p} - \frac{g_\eta(X)}{p\left(1 - g_\eta(X)\right)} \cdot (1-A) \cdot (Y - Q(0, X)) - A\tau^{\text{CDE}}, \quad (4.6)$$

where $p = P(A = 1)$. Then, the standard double machine learning estimator of $\tau^{\text{CDE}}$ Chernozhukov et al. (2016), and the $\alpha$-level confidence interval of this estimator, is given by

$$\hat{\tau}^{\text{TI}} = \frac{1}{n}\sum_{i=1}^n \hat{\phi}_i, \ CI^{\text{TI}} = \left(\hat{\tau}^{\text{TI}} - z_{1-\alpha/2}\hat{sd}(\hat{\phi}_i - A_i \cdot \hat{\tau}^{\text{TI}}/\hat{p}), \ \hat{\tau}^{\text{TI}} + z_{1-\alpha/2}\hat{sd}(\hat{\phi}_i - A_i \cdot \hat{\tau}^{\text{TI}}/\hat{p})\right),$$

$$(4.7)$$

where

$$\hat{\phi}_i = \frac{A_i \cdot \left(Y_i - \hat{Q}_0(X_i)\right)}{\hat{p}} - \frac{\hat{g}_\eta(X_i)}{\hat{p}\left(1 - \hat{g}_\eta(X_i)\right)} \cdot (1-A_i) \cdot \left(Y_i - \hat{Q}_0(X_i)\right), \ i = 1, \cdots, n, \quad (4.8)$$

$\hat{p} = \frac{1}{n}\sum_{i=1}^n A_i$, $z_{1-\alpha/2}$ is the $\alpha/2$-upper quantile of the standard normal, and $\hat{sd}(\cdot)$ is the sample standard deviation.

**Validity**   We now have an estimation procedure. It remains to give conditions under which this procedure is valid. In particular, we require that it should yield a consistent estimate and asymptotically correct confidence intervals.

**Theorem 2.** *Assume the following.*

1. *The mis-estimation of conditional outcomes can be bounded as follows*

$$\max_{a\in\{0,1\}} \mathbb{E}[(\hat{Q}_a(X)-Q(a,X))^2]^{\frac{1}{2}} = o(n^{-\frac{1}{4}}).  \tag{4.9}$$

2. *The propensity score function $P(A=1|\cdot,\cdot)$ is Lipschitz continuous on $\mathbb{R}^2$, and $\exists\ \varepsilon > 0$, $P\big(\varepsilon \le g_\eta(X) \le 1-\varepsilon\big) = 1$*

3. *The propensity score estimate converges at least as quickly as k nearest neighbor; i.e., $\mathbb{E}[\big(\hat{g}_\eta(X)-P(A=1\mid\hat{\eta}(X)\big)^2\mid\hat{\eta}(X)]^{\frac{1}{2}} = O(n^{-\frac{1}{4}})$ Györfi et al. (2002);*

4. *There exist positive constants $C_1$, $C_2$, $c$, and $q > 2$ such that*

$$\mathbb{E}[|Y|^q]^{\frac{1}{q}} \le C_2,\quad \sup_{\eta\in supp(\eta(X))} \mathbb{E}[(Y-Q(A,X)^2\mid\eta(X)=\eta)] \le C_2,$$

$$\mathbb{E}[(Y-Q(A,X)^2)]^{\frac{1}{2}} \ge c,\ \max_{a\in\{0,1\}} \mathbb{E}[\big|\hat{Q}_a(X)-Q(a,X)\big|]^{\frac{1}{q}} \le C_1.$$

*Then, the estimator $\hat{\tau}^{TI}$ is consistent and*

$$\sqrt{n}(\hat{\tau}^{TI}-\tau^{\text{CDE}}) \xrightarrow{d} \mathbb{N}(0,\sigma^2)  \tag{4.10}$$

*where $\sigma^2 = E\big(\phi(X;Q,g_\eta,\tau^{\text{CDE}})\big)^2$.*

The proof is provided in Appendix A.

The key point from this theorem is that we get asymptotic normality at the (fast) $\sqrt{n}$-rate while requiring only a slow ($n^{1/4}$) convergence rate of $Q$. Intuitively, the reason is simply that, because $\hat{\eta}(X)$ is only 2-dimensional, it is always possible to nonparametrically estimate the propensity score from $\hat{\eta}$ at a fast rate—even naive KNN works! Effectively, this means the rate at which we estimate the true propensity score $g_\eta(X) = P(A=1\mid\eta(X))$ is dominated by the rate at which we estimate $\eta(X)$, which is in turn determined by the rate for $\hat{Q}$. Now, the key property of the double ML estimator is that convergence only depends on the *product* of the convergence rates of $\hat{Q}$ and $\hat{g}$. Accordingly, this procedure is robust in the sense that we only need to estimate $\hat{Q}$ at the square root of the rate we needed for the naive $Q$-only procedure. This is much more plausible in practice. As we will see in Section 5, the TI-estimator dramatically improves the quality of the estimated confidence intervals and reduces the absolute bias of estimation.

*Remark* 3. In addition to robustness to noisy estimation of $Q$, there are some other advantages this estimation procedure inherits from the double ML estimator. If $\hat{Q}$ is consistent, then the estimator is nonparametrically efficient in the sense that no other non-parametric estimator has a smaller asymptotic variance. That is, the procedure using the data as efficiently as possible.

## 5   EXPERIMENTS

We empirically study the method's capability to provide accurate causal estimates with good uncertainty quantification Testing using semi-synthetic data (where ground truth causal effects are known), we find that the estimation procedure yields accurate causal estimates and confidence intervals. In particular, the TI-estimator has significantly lower absolute bias and vastly better uncertainty quantification than the $Q$-only method.

Additionally, we study the effect of the choice of nonparametric propensity score estimator and the choice of double machine-learning estimator, and the method's robustness in regard to $\hat{Q}$'s miscalibration. These results are reported in Appendices C and D. Although these

**Table 1:** The TI-estimator significantly improves both bias and coverage relative to the baseline. Tables show average absolute bias and confidence interval coverage of CDE estimates, over 100 resimulations. The TI estimator $\hat{\tau}^{\text{TI}}$ displays higher accuracy/smaller absolute bias of point estimate and much larger coverage proportions compared to outcome-only estimator $\hat{\tau}^{Q}$. The treatment level equals true CDE, which takes 1.0 (with causal effect) and 0.0 (without causal effect). Low and high noise level corresponds to $\gamma$ set to 1.0 and 4.0. Low and high confounding level corresponds to $\beta_c$ set to 50.0 and 100.0.

**(a)** Average absolute bias

| Noise: | Low | | | | High | | | |
|---|---|---|---|---|---|---|---|---|
| True CDE: | 1.0 | | 0.0 | | 1.0 | | 0.0 | |
| Confounding: | Low | High | Low | High | Low | High | Low | High |
| $\hat{\tau}^{Q}$ | 0.100 | 0.376 | 0.076 | 0.326 | 0.563 | 0.548 | 0.502 | 0.498 |
| $\hat{\tau}^{\text{TI}}$ | 0.069 | 0.059 | 0.114 | 0.074 | 0.088 | 0.049 | 0.002 | 0.089 |

**(b)** Coverage proportions of 95% confidence intervals

| Noise: | Low | | | | High | | | |
|---|---|---|---|---|---|---|---|---|
| True CDE: | 1.0 | | 0.0 | | 1.0 | | 0.0 | |
| Confounding: | Low | High | Low | High | Low | High | Low | High |
| $\hat{\tau}^{Q}$ | 0% | 0% | 2% | 0% | 0% | 0% | 0% | 0% |
| $\hat{\tau}^{\text{TI}}$ | 57% | 84% | 57% | 79% | 87% | 80% | 77% | 81% |

choices do not matter asymptotically, we find they have a significant impact in actual finite sample estimation. We find that, in general, kernel regression works well for propensity score estimation and the vanilla the Augmented Inverse Probability of Treatment weighted Estimator (AIPTW) corresponding to the CDE works well.

Finally, we reproduce the real-data analysis from Pryzant et al. (2020). We find that politeness has a positive effect on reducing email response time.

### 5.1 AMAZON REVIEWS

**Dataset** We closely follow the setup of Pryzant et al. (2020). We use publicly available Amazon reviews for music products as the basis for our semi-synthetic data. We include reviews for mp3, CD and vinyl, and among these exclude reviews for products costing more than \$100 or shorter than 5 words. The treatment $A$ is whether the review is five stars ($A = 1$) or one/two stars ($A = 0$).

To have a ground truth causal effect, we must now simulate the outcome. To produce a realistic dataset, we choose a real variable as the confounder. Namely, the confounder $C$ is whether the product is a CD ($C = 1$) or not ($C = 0$). Then, outcome $Y$ is generated according to $Y \leftarrow \beta_a A + \beta_c (\pi(C) - \beta_o) + \gamma N(0, 1)$. The true causal effect is controlled by $\beta_a$. We choose $\beta_a = 1.0, 0.0$ to generate data with and without causal effects. In this setting, $\beta_a$ is the oracle value of our causal estimand. The strength of confounding is controlled by $\beta_c$. We choose $\beta_c = 50.0, 100.0$. The ground-truth propensity score is $\pi(C) = P(A = 1|C)$. We set it to have the value $\pi(0) = 0.8$ and $\pi(1) = 0.6$ (by subsampling the data). $\beta_o$ is an offset $\mathbb{E}[\pi(C)] = \pi(0)P(C = 0) + \pi(1)P(C = 1)$, where $P(C = a)$, $a = 0, 1$ are estimated from data. Finally, the noise level is controlled by $\gamma$; we choose 1.0 and 4.0 to simulate data with small and large noise. The final dataset has $10,685$ data entries.

**Protocol** For the language model, we use the pretrained `distilbert-base-uncased` model provided by the `transformers` package. The model is trained in the k-folding fashion with 5 folds. We apply the Adam optimizer (Kingma & Ba, 2014) with a learning rate of $2e^{-5}$ and a batch size of 64. The maximum number of epochs is set as 20, with early stopping based on validation loss with a patience of 6. Each experiment is replicated with five different seeds and the final $\hat{Q}(a, x_i)$ predictions are obtained by averaging the predictions from the 5 resulting models. The propensity model is implemented by running the Gaussian process regression using `GaussianProcessClassifier` in the `sklearn` package with `DotProduct +` `WhiteKernel` kernel. (We choose different random state for the GPR to guarantee the convergence of the GPR.) The coverage experiment uses 100 replicates.

**Table 2:** Politeness has a positive causal effect on response time. The table displays different CDE estimates and their 95% confidence intervals. The unadjusted one is the difference of sample means of treatment (polite) group and control group. The confidence interval of $\hat{\tau}^{TI}$ only covers positive values, which means politeness can increase the probability of timely response.

| Estimator | CDE | Confidence Interval |
|---|---|---|
| *unadjusted* $\hat{\tau}^{naive}$ | -0.038 | [ -0.0679, -0.0073 ] |
| $\hat{\tau}^{Q}$ | 0.195 | [ 0.1910, 0.1993 ] |
| $\hat{\tau}^{TI}$ | 0.200 | [ 0.1708, 0.2288 ] |

**Results**   The main question here is the efficacy of the estimation procedure. Table 1 compares the outcome-only estimator $\hat{\tau}^{Q}$ and the estimator $\hat{\tau}^{TI}$. First, the absolute bias of the new method is significantly lower than the absolute bias of the outcome-only estimator. This is particularly true where there is moderate to high levels of confounding. Next, we check actual coverage rates over 100 replicates of the experiment. First, we find that the naive approach for the outcome-only estimator fails completely. The nominal confidence interval almost never actually includes the true effect. It is wildly optimistic. By contrast, the confidence intervals from the new method often cover the true value. This is an enormous improvement over the baseline. Nevertheless, they still do not actually achieve their nominal (95%) coverage. This may be because the $\hat{Q}$ estimate is still not good enough for the asymptotics to kick in, and we are not yet justified in ignoring the uncertainty from model fitting.

### 5.2 Application: Consumer Complaints to the Financial Protection Bureau

We follow the same pipeline of the real data experiment in (Pryzant et al., 2020, §6.2). The dataset is consumers complaints made to the financial protection. Treatment $A$ is politeness (measured using Yeomans et al. (2018)) and the outcome $Y$ is a binary indicator of whether complaints receive a response within 15 days.

We use the same training procedure as for the simulation data. Table 2 shows point estimates and their 95% confidence intervals. Notice that the naive estimator show a significant *negative* effect of politeness on reducing response time. On the other hand, the more accurate AIPTW method as well as the outcome-only estimator have confidence intervals that cover only positive values, so we conclude that consumers' politeness has a positive effect on response time. This matches our intuitions that being more polite should increase the probability of receiving a timely reply.

## 6 Discussion

In this paper, we address the estimation of the causal effect of a text document attribute using observational data. The key challenge is that we must adjust for the text—to handle confounding—but adjusting for all of the text violates overlap. We saw that this issue could be effectively circumvented with a suitable choice of estimand and estimation procedure. In particular, we have seen an estimand that corresponds to the qualitative causal question, and an estimator that is valid even when the outcome model is learned slowly. The procedure also circumvents the need for bootstrapping, which is prohibitively expensive in our setting.

There are some limitations. The actual coverage proportion of our estimator is below the nominal level. This is presumably due to the imperfect fit of the conditional outcome model. Diagnostics (see Appendix D) show that as conditional outcome estimations become more accurate, the TI estimator becomes less biased, and its coverage increases. It seems plausible that the issue could be resolved by using more powerful language models.

Although we have focused on text in this paper, the problem of causal estimation with apparent overlap violation exists in any problem where we must adjust for unstructured and high-dimensional covariates. Another interesting direction for future work is to understand how analogous procedures work outside the text setting.

ACKNOWLEDGEMENT

Thanks to Alexander D'Amour for feedback on an earlier draft. We acknowledge the University of Chicago's Research Computing Center for providing computing resources. This work was partially supported by Open Philanthropy.

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

## A  PROOF OF ASYMPTOTIC NORMALITY

**Theorem 2.** *Assume the following.*

1. *The mis-estimation of conditional outcomes can be bounded as follows*

$$\max_{a \in \{0,1\}} \mathbb{E}[(\hat{Q}_a(X) - Q(a,X))^2]^{\frac{1}{2}} = o(n^{-\frac{1}{4}}). \tag{4.9}$$

2. *The propensity score function $P(A = 1|\cdot,\cdot)$ is Lipschitz continuous on $\mathbb{R}^2$, and $\exists\ \varepsilon > 0$, $P\left(\varepsilon \le g_\eta(X) \le 1 - \varepsilon\right) = 1$*

3. *The propensity score estimate converges at least as quickly as k nearest neighbor; i.e., $\mathbb{E}[\left(\hat{g}_\eta(X) - P(A = 1 \mid \hat{\eta}(X))\right)^2 \mid \hat{\eta}(X)]^{\frac{1}{2}} = O(n^{-\frac{1}{4}})$ Györfi et al. (2002);*

4. *There exist positive constants $C_1$, $C_2$, $c$, and $q > 2$ such that*

$$\mathbb{E}[|Y|^q]^{\frac{1}{q}} \le C_2, \quad \sup_{\eta \in supp(\eta(X))} \mathbb{E}[(Y - Q(A,X)^2 \mid \eta(X) = \eta)] \le C_2,$$

$$\mathbb{E}[(Y - Q(A,X)^2)]^{\frac{1}{2}} \ge c, \quad \max_{a \in \{0,1\}} \mathbb{E}[\left|\hat{Q}_a(X) - Q(a,X)\right|]^{\frac{1}{q}} \le C_1.$$

*Then, the estimator $\hat{\tau}^{TI}$ is consistent and*

$$\sqrt{n}(\hat{\tau}^{TI} - \tau^{\text{CDE}}) \xrightarrow{d} \mathbb{N}(0, \sigma^2) \tag{4.10}$$

*where $\sigma^2 = E\left(\phi(X; Q, g_\eta, \tau^{\text{CDE}})\right)^2$.*

*Proof.* We first prove that misestimation of propensity score has the rate $n^{-\frac{1}{4}}$. For simplicity, we use $f_g$, $\hat{f}_g$: $(u,v) \in \mathbb{R}^2 \to \mathbb{R}$ to denote conditional probability $P(A = 1|u,v) = f_g(u,v)$ and the estimated propensity function by running the nonparametric regression $\hat{P}(A = 1|u,v) = \hat{f}_g(u,v)$. Specifically, we have $f_g(Q(0,X), Q(1,X)) = g_\eta(X)$ and $\hat{f}_g(\hat{Q}_0(X), \hat{Q}_1(X)) = \hat{P}(A = 1|\hat{Q}_0(X), \hat{Q}_1(X)) = \hat{g}_\eta(X)$. Since $\mathbb{E}[\left(\hat{Q}_0(X) - Q(0,X)\right)^2]^{\frac{1}{2}}$, $\mathbb{E}[\left(\hat{Q}_1(X) - Q(1,X)\right)^2]^{\frac{1}{2}} = o(n^{-1/4})$ and $f_g$ is Lipschitz continuous, we have

$$\mathbb{E}\left[\left|\left|f_g(\hat{Q}_0(X), \hat{Q}_1(X)) - f_g(Q(0,X), Q(1,X))\right|\right|^2\right]^{\frac{1}{2}}$$

$$\le L \cdot \mathbb{E}\left[\left|\left|\left(\hat{Q}_0(X), \hat{Q}_1(X)\right) - (Q(0,X), Q(1,X))\right|\right|_2^2\right]^{\frac{1}{2}} \tag{A.1}$$

$$= L \cdot \left\{\mathbb{E}\left[\left(\hat{Q}_0(X) - Q(0,X)\right)^2\right] + \mathbb{E}\left[\left(\hat{Q}_1(X) - Q(1,X)\right)^2\right]\right\}^{\frac{1}{2}}$$

$$= o(n^{-1/4})$$

Since the true propensity function $f_g$ is Lipschitz continuous on $\mathbb{R}^2$, the mean squared error rate of the k nearest neighbor is $O(n^{-1/2})$ Györfi et al. (2002). In addition, since the propensity score function and its estimation are bounded under 1, we have the following equation

$$\mathbb{E}\left|\hat{f}_g(\hat{Q}_0(X), \hat{Q}_1(X)) - f_g(\hat{Q}_0(X), \hat{Q}_1(X))\right|^2 = O(n^{-1/2}), \tag{A.2}$$

due to the dominated convergence theorem. By (A.1) and (A.2), we can bound the mean squared error of estimated propensity score in the following form:

$$
\begin{aligned}
&\mathbb{E}\left[\left(\hat{g}_\eta(X) - g_\eta(X)\right)^2\right] \\
&\leq \mathbb{E}\left[\left(\hat{g}_\eta(X) - f_g(\hat{Q}_0(X), \hat{Q}_1(X))\right)^2\right] + \mathbb{E}\left[\left(f_g(\hat{Q}_0(X), \hat{Q}_1(X)) - g_\eta(X)\right)^2\right] \\
&= \mathbb{E}\left|f_g(\hat{Q}_0(X), \hat{Q}_1(X)) - f_g(Q(0,X), Q(1,X))\right|^2 + \\
&\quad \mathbb{E}\left|\hat{f}_g(\hat{Q}_0(X), \hat{Q}_1(X)) - f_g(\hat{Q}_0(X), \hat{Q}_1(X))\right|^2 \\
&= O(n^{-1/2}),
\end{aligned}
\tag{A.3}
$$

that is $\mathbb{E}\left[\left(\hat{g}_\eta(X) - g_\eta(X)\right)^2\right]^{\frac{1}{2}} = O(n^{-\frac{1}{4}})$.

Before we apply the conclusion of Theorem 5.1 in (Chernozhukov et al., 2017b), we need to check all assumptions in Assumption 5.1 hold in Chernozhukov et al. (2017b). Let $C := \max\left\{(2C_1^q + 2^q)^{\frac{1}{q}}, C_2\right\}$.

(a) $\mathbb{E}[Y - Q(A,X) \mid \eta(X), A] = 0$, $\mathbb{E}[A - g_\eta(X) \mid \eta(X)] = 0$ are easily checked by invoking definitions of $Q$ and $g_\eta$.

(b) $\mathbb{E}[|Y|^q]^{\frac{1}{q}} \leq C$, $\mathbb{E}[(Y - Q(A,X))^2]^{\frac{1}{2}} \geq c$, and $\sup_{\eta \in \mathrm{supp}(\eta(X))} \mathbb{E}[(Y - Q(A,X))^2 \mid \eta(X) = \eta] \leq C$ are guaranteed by the fourth condition in the theorem.

(c) $\mathrm{P}\left(\varepsilon \leq g_\eta(X) \leq 1 - \varepsilon\right) = 1$ is the second condition in the theorem.

(d) Since propensity score function and its estimation are bounded under 1, we have

$$
\begin{aligned}
&\left(\mathbb{E}[|\hat{Q}_1(X) - Q(1,X)|^q] + \mathbb{E}[|\hat{Q}_0(X) - Q(0,X)|^q] + \mathbb{E}[|\hat{g}_\eta(X) - g_\eta(X)|^q]\right)^{\frac{1}{q}} \\
&\leq (C_1^q + C_1^q + 2^q)^{\frac{1}{q}} \\
&\leq C
\end{aligned}
$$

(e) Based on (A.3) and condition 1 in the theorem, we have

$$
\begin{aligned}
&\left(\mathbb{E}[(\hat{Q}_1(X) - Q(1,X))^2] + \mathbb{E}[(\hat{Q}_0(X) - Q(0,X))^2] + \mathbb{E}[(\hat{g}_\eta(X) - g_\eta(X))^2]\right)^{\frac{1}{2}} \\
&\leq \left[o(n^{-\frac{1}{2}}) + o(n^{-\frac{1}{2}}) + O(n^{-\frac{1}{2}})\right]^{\frac{1}{2}} \\
&\leq O(n^{-\frac{1}{4}}), \\
&\mathbb{E}[(\hat{Q}_0(X) - Q(0,X))^2]^{\frac{1}{2}} \cdot \mathbb{E}[(\hat{g}_\eta(X) - g_\eta(X))^2]^{\frac{1}{2}} = o(n^{-\frac{1}{2}})
\end{aligned}
$$

(f) Based on condition 3 in the theorem, we have

$$
\sup_{x \in \mathrm{supp}(X)} \mathbb{E}[(\hat{g}_\eta(X) - \mathrm{P}(A = 1 \mid \hat{\eta}(X)))^2 \mid \hat{\eta}(X) = \hat{\eta}(x)] = O(n^{-\frac{1}{2}}).
$$

We consider a smaller positive constant $\tilde{\varepsilon}$ instead of $\varepsilon$. Note that for $\tilde{\varepsilon} < \varepsilon$, we still have $P(\tilde{\varepsilon} \leq g_\eta(X) \leq 1 - \tilde{\varepsilon}) = 1$. Then,

$$
P\left( \sup_{x \in \text{supp}(X)} \left| \hat{g}_\eta(x) - \frac{1}{2} \right| > \frac{1}{2} - \tilde{\varepsilon} \right) = P\left( \inf_{x \in \text{supp}(X)} \hat{g}_\eta(x) < \tilde{\varepsilon} \right) + P\left( \sup_{x \in \text{supp}(X)} \hat{g}_\eta(x) > 1 - \tilde{\varepsilon} \right)
$$

$$
\leq P\left( \inf_{x \in \text{supp}(X)} P(A = 1 \mid \hat{\eta}(X) = \hat{\eta}(x)) - \inf_{x \in \text{supp}(X)} \hat{g}_\eta(x) > \varepsilon - \tilde{\varepsilon} \right)
$$

$$
+ P\left( \sup_{x \in \text{supp}(X)} \hat{g}_\eta(x) - \sup_{x \in \text{supp}(X)} P(A = 1 \mid \hat{\eta}(X) = \hat{\eta}(x)) > 1 - \tilde{\varepsilon} - (1 - \varepsilon) \right)
$$

$$
\leq \frac{\mathbb{E}\left[ \left( \inf_{x \in \text{supp}(X)} \hat{g}_\eta(x) - \inf_{x \in \text{supp}(X)} P(A = 1 \mid \hat{\eta}(X) = \hat{\eta}(x)) \right)^2 \right]}{(\varepsilon - \tilde{\varepsilon})^2} +
$$

$$
\frac{\mathbb{E}\left[ \left( \sup_{x \in \text{supp}(X)} \hat{g}_\eta(x) - \sup_{x \in \text{supp}(X)} P(A = 1 \mid \hat{\eta}(X) = \hat{\eta}(x)) \right)^2 \right]}{(\varepsilon - \tilde{\varepsilon})^2}
$$

$$
\leq \frac{2 \sup_{x \in \text{supp}(X)} \mathbb{E}\left[ \left( \hat{g}_\eta(X) - P(A = 1 \mid \hat{\eta}(X) = \hat{\eta}(x)) \right)^2 \right]}{(\varepsilon - \tilde{\varepsilon})^2}
$$

$$
= O(n^{-\frac{1}{2}})
$$

Hence, $P(\sup_{x \in \text{supp}(X)} \left| \hat{g}_\eta(x) - \frac{1}{2} \right| \leq \frac{1}{2} - \tilde{\varepsilon}) \geq 1 - O(n^{-\frac{1}{2}})$.

With (a)-(f), we can invoke the conclusion in Theorem 5.1 in (Chernozhukov et al., 2017b), and get the asymptotic normality of the TI estimator. □

## B   PROOF OF CAUSAL IDENTIFICATION

**Theorem 1.** *Assume the following:*
*1. (Causal structure) The causal relationships among A, $\tilde{A}$, Z, Y, and X satisfy the causal DAG in Figure 2;*
*2. (Overlap) $0 < P(A = 1 \mid X_{A \wedge Z}, X_Z) < 1$;*
*3. (Intention equals perception) $A = \tilde{A}$ almost surely with respect to all interventional distributions. Then, the CDE is identified from observational data as*

$$
\text{CDE} = \tau^{\text{CDE}} := \mathbb{E}_{X \mid \tilde{A} = 1}\left[ \mathbb{E}[Y \mid \eta(X), \tilde{A} = 1] - \mathbb{E}[Y \mid \eta(X), \tilde{A} = 0] \right], \tag{3.4}
$$

*where $\eta(X) := (Q(0, X), Q(1, X))$.*

*Proof.* We first prove that this two-dimensional confounding part $\eta(X)$ satisfies positivity. Since $(Q(0, X), Q(1, X)) = (\mathbb{E}[Y \mid A = 1, X_{A \wedge Z}, X_Z], \mathbb{E}[Y \mid A = 0, X_{A \wedge Z}, X_Z])$ is a function of $(X_{A \wedge Z}, X_Z)$, the following equations hold:

$$
\begin{aligned}
P(A = 1 \mid Q(0, X), Q(1, X)) &= \mathbb{E}(A \mid Q(0, X), Q(1, X)) \\
&= \mathbb{E}[E(A \mid X_{A \wedge Z}, X_Z) \mid Q(0, X), Q(1, X)] \\
&= \mathbb{E}[P(A = 1 \mid X_{A \wedge Z}, X_Z) \mid Q(0, X), Q(1, X)].
\end{aligned} \tag{B.1}
$$

As $0 < P(A = 1 \mid X_{A \wedge Z}, X_Z) < 1$, we have $0 < P(A = 1 \mid Q(0, X), Q(1, X)) < 1$. Furthermore, we have $0 < P(\tilde{A} = 1 \mid Q(0, X), Q(1, X)) < 1$ due to almost everywhere equivalence of $A$ and $\tilde{A}$.

Since $A = \tilde{A}$, we can rewrite (3.1) by replacing $A$ with $\tilde{A}$ in the following form:

$$
\begin{aligned}
\text{CDE} &= \mathbb{E}_{X_{A \wedge Z}, X_Z \mid \tilde{A}=1} \big[ \; \mathbb{E}(Y \mid \text{do}(\tilde{A}=1), X_{A \wedge Z}, X_Z) - \mathbb{E}(Y \mid \text{do}(\tilde{A}=0), X_{A \wedge Z}, X_Z) \; \big] \\
&= \mathbb{E}_{X_{A \wedge Z}, X_Z \mid \tilde{A}=1} \big[ \; \mathbb{E}(Y \mid \tilde{A}=1, X_{A \wedge Z}, X_Z) - \mathbb{E}(Y \mid \tilde{A}=0, X_{A \wedge Z}, X_Z) \; \big] \\
&= \mathbb{E}_{X_{A \wedge Z}, X_Z \mid \tilde{A}=1} \big[ \; \mathbb{E}(Y \mid \tilde{A}=1, X) - \mathbb{E}(Y \mid \tilde{A}=0, X) \; \big] \\
&= \mathbb{E}_{X_{A \wedge Z}, X_Z \mid \tilde{A}=1} \big[ \mathbb{E}(Y \mid \tilde{A}=1, Q(0,X), Q(1,X)) \big] - \mathbb{E} \big[ \mathbb{E}(Y \mid \tilde{A}=0, Q(0,X), Q(1,X)) \big] \\
&= \mathbb{E}_{X_{A \wedge Z}, X_Z \mid \tilde{A}=1} \big[ \mathbb{E}(Y \mid \tilde{A}=1, \eta(X)) \big] - \mathbb{E} \big[ \mathbb{E}(Y \mid \tilde{A}=0, \eta(X)) \big] \\
&= \mathbb{E}_{X \mid \tilde{A}=1} \big[ \mathbb{E}(Y \mid \tilde{A}=1, \eta(X)) \big] - \mathbb{E} \big[ \mathbb{E}(Y \mid \tilde{A}=0, \eta(X)) \big].
\end{aligned}
\tag{B.2}
$$

The equivalence of the first and the second line is because $X_{A \wedge Z}$, $X_Z$ block all backdoor paths between $\tilde{A}$ and $Y$ (See Figure 2) and $0 < \text{P}(\tilde{A}=1 \mid Q(0,X), Q(1,X)) < 1$. Thus, the "*do-operation*" in the first line can be safely removed. Equivalence of the second line and the third line is due to $Q(\tilde{A}, X) = \mathbb{E}\big(Y \mid \tilde{A}, X_{A \wedge Z}, X_Z\big)$, which is subject to the causal model in Figure 2. The last equation is based on the fact that $\eta(X)$ is a function of only $X_{A \wedge Z}$ and $X_Z$. (It can be easily checked by using the definition of the expectation.)

(B.2) shows that $(Q(0,X), Q(1,X))$ is a two-dimensional confounding variable such that CDE is identifiable when we adjust for it as the confounding part.

$\square$

Note that if $f$ and $h$ are two invertible functions on $\mathbb{R}$, $(f(Q(0,X)), h(Q(1,X)))$ also suffices the identification for CDE. Since the sigma algebra should be the same for $(Q(0,X), Q(1,X))$ and $f(Q(0,X)), h(Q(1,X))$, i.e.,

$$
\sigma\left(Q(0,X), Q(1,X)\right) = \sigma\left(f(Q(0,X)), h(Q(1,X))\right).
$$

Hence, we have

$$
\begin{aligned}
&\text{P}\left(A=1 \mid Q(0,X), Q(1,X)\right) = \text{P}\left(A=1 \mid f(Q(0,X)), h(Q(1,X))\right), \\
&\mathbb{E}\left(Y \mid Q(0,X), Q(1,X)\right) = \mathbb{E}\left(Y \mid f(Q(0,X)), h(Q(1,X))\right).
\end{aligned}
\tag{B.3}
$$

## C  ADDITIONAL EXPERIMENTS

We conduct additional experiments to show how the estimation of causal effect changes 1) over different nonparametric models for the propensity score estimation, and 2) when using different double machine learning estimators on causal estimation. Specifically, for the first study, we apply different nonparametric models and the logistic regression to the estimated confounding part $\hat{\eta}(X) = \big(\hat{Q}_0(X), \hat{Q}_1(X)\big)$ to obtain propensity scores. We use ATT AIPTW in all above cases for causal effect estimation. For the second study, we fix the first two stages of the TI estimator, i.e. we apply Q-Net for the conditional outcomes and compute propensity scores with the Gaussian process regression where the kernel function is the summation of dot product and white noise. Estimated conditional outcomes and propensity scores are plugged into different double machine learning estimators. We make the following conclusions with results of above experiments.

**The choice of nonparametric models is significant.**   Table 3 summarizes results with applying different regression models for the propensity estimation. We can see that suitable nonparametric models will strongly increase the coverage proportion over true causal estimand. Therefore, we conclude that the accuracy in causal estimation is highly dependent on the choice of nonparametric models. In practice, when there is some prior information about the propensity score function, we should apply the most suitable nonparametric model to increase the reliability of our causal estimation.

**The ATT AIPTW is consistently the best double machine learning estimator.**   Table 4 shows results by applying different double machine learning estimators. We apply both estimators for the average treatment effect (ATE) and the controlled direct effect (CDE). The bias of "unadjusted" estimator $\hat{\tau}^{\text{naive}}$ is also included in Table 4 (a). For absolute bias,

ATT AIPTW $\hat{\tau}^{\text{TI}}$ has comparable results with other double machine learning estimators in most cases. For coverage proportion of confidence intervals, though it has lower rates in some cases, $\hat{\tau}^{\text{TI}}$ has consistently the best performance. Especially in high confounding situations, the advantage of $\hat{\tau}^{\text{TI}}$ is obvious.

**Estimator** For each dataset, we compute estimators as follows. $n_1$ and $n_0$ stands for the number of individuals in the treated and controlled group. $n = n_1 + n_0$ is the total number of individuals.

- "Unadjusted" baseline estimator: $\hat{\tau}^{\text{naive}} = \frac{1}{n_1}\sum_{i:A_i=1} Y_i - \frac{1}{n_0}\sum_{i:A_i=0} Y_i$

- "Outcome-only" estimator: $\hat{\tau}^{\text{Q}} = \frac{1}{n_1}\sum_{i:A_i=1}\hat{Q}_{1,i} - \hat{Q}_{0,i}$

- ATT AIPTW: $\hat{\tau}^{\text{TI}} = \frac{1}{n_1}\sum_{i:A_i=1} A_i(Y_i - \hat{Q}_{0,i}) - (1-A_i)(Y_i - \hat{Q}_{0,i})\frac{\hat{g}_i}{1-\hat{g}_i}$

**Table 3:** The choice of nonparametric models for the TI-estimator is significant. Tables show average absolute bias and 95% confidence intervals' coverage of $\hat{\tau}^{\text{TI}}$ with applying different nonparametric models in the second stage. The Gaussian process regression with the dot product+ white noise kernel has the best performance (lowest absolute bias and highest coverage proportion). The treatment level is equal to true CDE, which takes 1.0 (with causal effect) and 0.0 (without causal effect). Low and high noise level corresponds to $\gamma = 1.0$ and 4.0. Low and high confounding level corresponds to $\beta_c = 50.0$ and 100.0.

**(a)** Average absolute bias

| Noise: | Low | | | | High | | | |
|---|---|---|---|---|---|---|---|---|
| Treatment (oracle causal effect): | 1.0 | | 0.0 | | 1.0 | | 0.0 | |
| Confounding: | Low | High | Low | High | Low | High | Low | High |
| GPR (Dot Product+White Noise) | 0.069 | 0.059 | 0.113 | 0.074 | 0.088 | 0.049 | 0.002 | 0.089 |
| GPR (RBF) | 0.150 | 0.348 | 0.156 | 0.329 | 0.363 | 0.452 | 0.344 | 0.424 |
| KNN | 0.147 | 0.334 | 0.144 | 0.313 | 0.316 | 0.372 | 0.304 | 0.356 |
| AdaBoost | 0.074 | 0.349 | 0.061 | 0.323 | 0.526 | 0.497 | 0.479 | 0.464 |
| Logistic | 0.070 | 0.057 | 0.114 | 0.073 | 0.086 | 0.047 | -0.001 | 0.087 |

**(b)** Coverage proportions of 95% confidence intervals

| Noise: | Low | | | | High | | | |
|---|---|---|---|---|---|---|---|---|
| Treatment (oracle causal effect): | 1.0 | | 0.0 | | 1.0 | | 0.0 | |
| Confounding: | Low | High | Low | High | Low | High | Low | High |
| GPR (Dot Product+White Noise) | 57% | 84% | 57% | 79% | 87% | 80% | 77% | 81% |
| GPR (RBF) | 31% | 0% | 41% | 0% | 7% | 7% | 17% | 19% |
| KNN | 18% | 0% | 39% | 0% | 11% | 8% | 11% | 8% |
| AdaBoost | 25% | 0% | 35% | 0% | 0% | 0% | 0% | 0% |
| Logistic | 58% | 84% | 57% | 79% | 87% | 80% | 78% | 81% |

# D DISCUSSION OF LOW COVERAGE

In this section, we discuss why the confidence intervals we get (See Table 1) have lower coverage than the nominated level 95%. We conduct diagnostics and find that the inaccuracy of $Q$'s estimations is responsible for the low coverage. We compute absolute biases, variances, and coverages of $\tau^{\text{TI}}$'s with different mean squared errors $\hat{\mathbb{E}}[(Q-\hat{Q})^2]$ by using different numbers of datasets. According to Figure 4–Figure 5, as the mean squared error of $Q$ increases, the bias of $\tau^{\text{TI}}$ grows and the coverage of $\tau^{\text{TI}}$ drops. Specifically, the highest coverage of each setting is almost 95% (use 50 datasets with most accurate conditional outcome estimations). In practice, one direct way to improve the TI estimator's accuracy is to apply better NLP models so that more accurate conditional outcome estimations can be obtained.

**Table 4:** The ATT AIPTW is consistently the best double machine learning estimator for this causal problem. Tables show average absolute bias and 95% confidence intervals' coverage of different causal estimations. ATT AIPTW $\hat{\tau}^{TI}$ shows consistently the lowest absolute bias and highest coverage rate. For propensity score estimation, the Gaussian process regression with the dot product+ white noise kernel is applied for all estimators. The treatment level is equal to true CDE/true ATE, which takes 1.0 (with causal effect) and 0.0 (without causal effect). Low and high noise level corresponds to $\gamma = 1.0$ and 4.0. Low and high confounding level corresponds to $\beta_c = 50.0$ and 100.0.

**(a)** Average absolute bias

| Noise: | Low | | | | High | | | |
|---|---|---|---|---|---|---|---|---|
| Treatment (oracle CDE): | 1.0 | | 0.0 | | 1.0 | | 0.0 | |
| Confounding: | Low | High | Low | High | Low | High | Low | High |
| unadjusted $\hat{\tau}^{naïve}$ | 1.071 | 2.143 | 1.071 | 2.1453 | 1.068 | 2.140 | 1.069 | 2.140 |
| ATE AIPTW | 0.094 | 0.178 | 0.128 | 0.195 | 0.122 | 0.106 | 0.061 | 0.140 |
| ATE BMM | 0.094 | 0.176 | 0.128 | 0.193 | 0.122 | 0.106 | 0.061 | 0.140 |
| ATE IPTW | -0.574 | -1.492 | -1.839 | -1.807 | -0.082 | -0.592 | -0.393 | -0.649 |
| ATT AIPTW: $\hat{\tau}^{TI}$ | 0.069 | 0.059 | 0.114 | 0.074 | 0.088 | 0.049 | 0.002 | 0.089 |
| ATT BMM | 0.075 | 0.147 | -0.031 | 0.062 | 0.621 | 0.454 | 0.464 | 0.337 |
| ATT TMLE: | 0.084 | 0.194 | 0.085 | 0.196 | 0.186 | 0.136 | 0.174 | 0.163 |

**(b)** Coverage Proportions of 95% confidence intervals

| Noise: | Low | | | | High | | | |
|---|---|---|---|---|---|---|---|---|
| Treatment (oracle CDE): | 1.0 | | 0.0 | | 1.0 | | 0.0 | |
| Confounding: | Low | High | Low | High | Low | High | Low | High |
| ATE AIPTW | 37% | 36% | 69% | 33% | 75% | 79% | 79% | 71% |
| ATE BMM | 39% | 35% | 70% | 36% | 75% | 79% | 79% | 71% |
| ATE IPTW | 11% | 1% | 0% | 1% | 90% | 39% | 44% | 37% |
| ATT AIPTW: $\hat{\tau}^{TI}$ | 57% | 84% | 57% | 79% | 87% | 80% | 77% | 81% |
| ATT BMM | 26% | 4% | 49% | 41% | 1% | 3% | 1% | 14% |
| ATT TMLE | 48% | 22% | 75% | 24% | 51% | 77% | 72% | 67% |

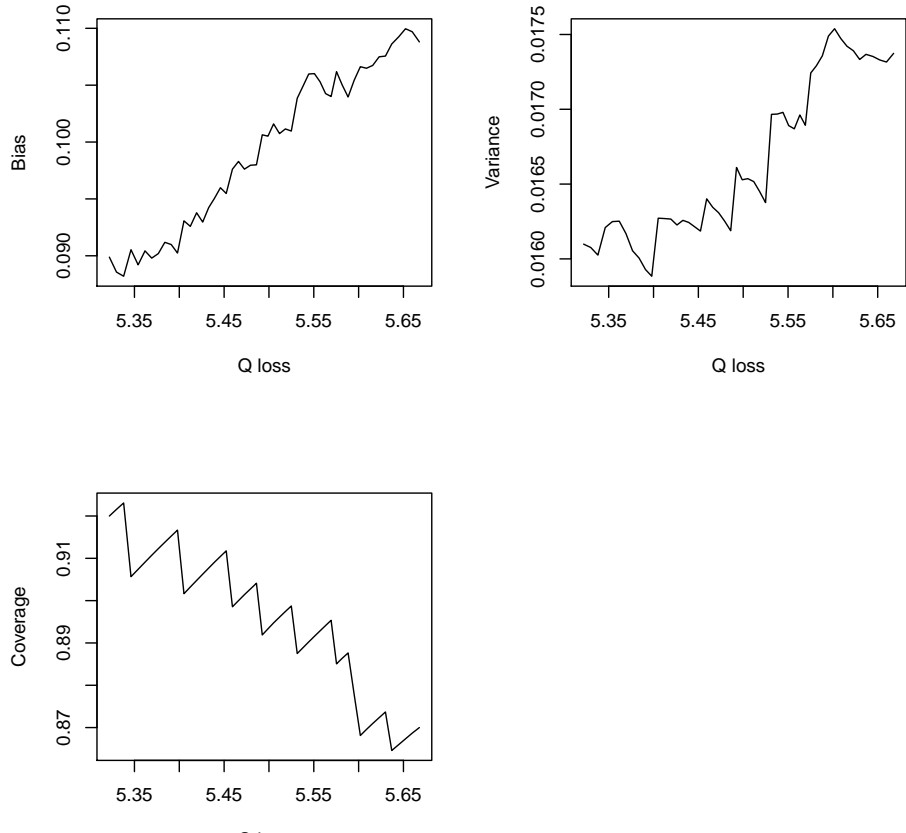

**Figure 4:** Absolute biases and variances increase while coverages decrease as the mean squared errors of $Q$ (Q loss) becomes larger. This experiment uses 100 datasets with $\beta_t = 1$ (with causal effect), $\beta_c = 50.0$ (low confounding), and $\gamma = 4.0$ (high noise).

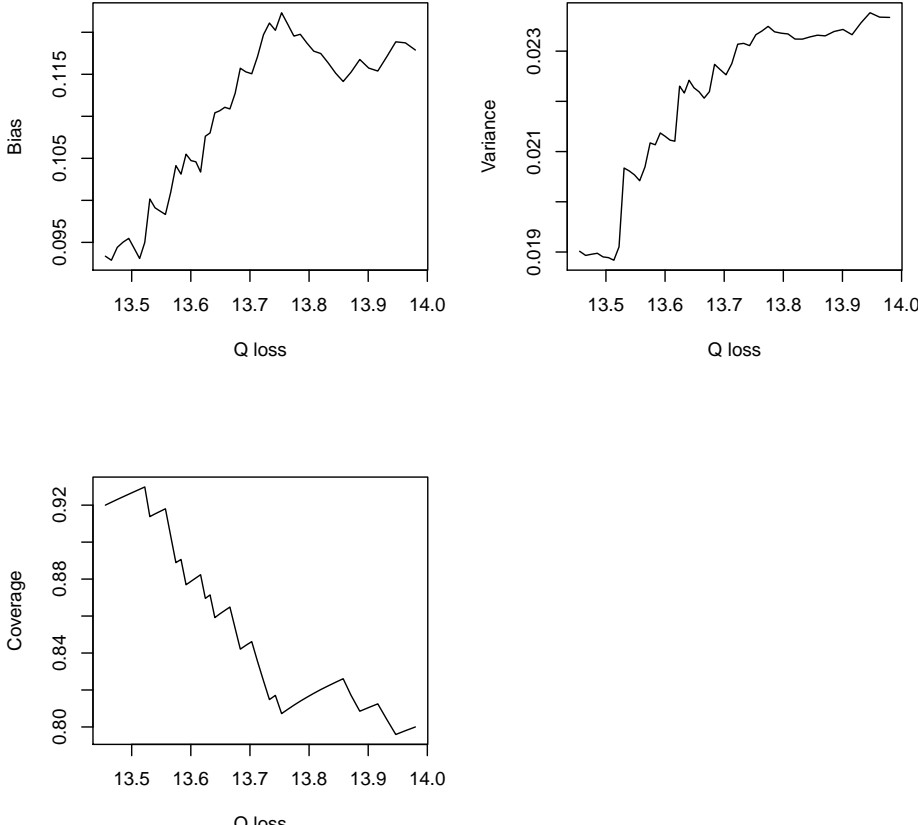

**Figure 5:** Absolute biases and variances increase while coverages decrease as the mean squared errors of $Q$ becomes larger. This experiment uses 100 datasets with $\beta_t = 1$ (with causal effect), $\beta_c = 100.0$ (high confounding), and $\gamma = 4.0$ (high noise).

