# OpenReview forum: "Causal Estimation for Text Data with (Apparent) Overlap Violations"
_ICLR.cc/2023/Conference — ICLR 2023 poster_

### Official Review · Reviewer_yJuA · 2022-10-21

**Confidence:** 4
**Correctness:** 4
**Technical Novelty And Significance:** 3
**Empirical Novelty And Significance:** 3
**Recommendation:** 6

**Clarity, Quality, Novelty And Reproducibility:**

The paper is well-written, technically sound, and provides theoretical justifications. Its clarity can be improved in a couple of places. The authors should make a more compelling case for the paper's novelty (both in methods and theory).


**Strength And Weaknesses:**

+1. the paper is well-written and technically correct.
+2. there are theoretical justifications for the proposed methods.
+3. the proposed method is backed up by simulations and two real data applications (Amazon reviews and consumer complaints.)
+4. the topic of the paper is increasingly important and the paper is very timely.

-1. there seems to be some small gap between the advocated main idea in Figure 2 (namely, one should eliminate the components in X that are predictive of A only) and the proposed method, which does not specifically make an effort to eliminate these components. Specifically, the authors proposed to use a three-head dragonnet where two heads predict Q0 and Q1 and one head predicts A. In the end, only the estimated Q0 and Q1 are used for making the causal effect estimation. Should I understand that the estimated A in the dragonnet represents the components in X predictive of A? If that's the case, how can we ensure that the two heads for Q0 and Q1 do not use information in X_A?  If not, how should we make the connection between the conceptual idea in Figure 2 and the implementation in Figure 3?
-2. related to (-1) above, it seems that a critical point is that Equation (3.3) Q(\tilde A, X) := E(Y | \tilde A,X) is in fact E (Y | \tilde A, X_A∧Z, X_Z). This seems to suggest that although conceptually CDE as in (3.2) should be based on X_A∧Z, X_Z which are unknown, in reality, one only needs to build models for Q(A,X) using the entire X. In this case, how does the proposed method differ from the naive method that "naively adjust for all the text as the confounding part"? Some clarification is needed.
-3. It would be helpful to add the "naively adjust for all the text" method to the simulation, to see how this method is biased and problematic.
-4. Clarification needed: following Theorem 1, the author said "We emphasize that, regardless of whether the overlap condition holds or not, the propensity score of η(X) in condition 2 is accessible and meaningful." There is no η(X) in condition 2, and since η(X) is the outcome model, it does not have a propensity score.
-5. abbreviation: On page 7, it says "AIPTW is double robust". AIPTW was not previously defined. AIPTW = augmented inverse probability T??? (treatment?) weighting? If you mean the estimator in (4.8), it should be said so.
-6. bias: about Table 1, you said "bias of the new method is significantly lower than the bias of the outcome-only estimator". Bias can be both positive and negative, and hence lower bias is not necessarily better. Do you in fact mean average estimation error or root mean square estimation error? In fact, it would be good for you to define the bias that you are reporting in the paper.
-7. simulation. It is counter-intuitive that low confounding actually leads to a low coverage rate of the confidence intervals. In randomized trials (zero confounding), shouldn't the confidence interval be valid? Some explanation/discussion is needed.
-8. Please explain the specific differences between the current work and Claudia Shi, David M. Blei, and Victor Veitch. Adapting neural networks for the estimation of treatment effects. In Advances in Neural Information Processing Systems, 2019.
-9. I am not sure how novel is the proof of Theorem 2. The stated results seem to inherit from double machine learning in Chernozhukov's work. The author should comment on how their proof differs from that of Chernozhukov. This is important as this part is claimed to be the second main contribution of the paper.


**Summary Of The Paper:**

This paper concerns causal effect estimation when the confounders and treatment are both derived from texts. One issue herein is that when adjusting for the entire text, there is a violation of the overlap assumption which is required for drawing valid causal conclusions. The proposed solution is to use supervised representation learning to produce a data representation that preserves confounding information while eliminating information that is only predictive of the treatment.

**Summary Of The Review:**

This paper analyzes the identification conditions for causal effect estimation when there is an apparent violation of overlap. The paper then proposed to use a neural network to learn a representation of the text to predict the outcome, which is then fed into the common double machine learning method to estimate the average treatment effect for the treated. There is a seeming gap between the conceptual idea and the implementation (which should be addressed after some revision/clarification). The author should also address how their paper is novel compared to previous methods (Shi et al.) and theory (that of double machine learning).

---

> ### Author Response · Authors · 2022-11-11
> **Authors' response**
>
> 1. *there seems to be some small gap between the advocated main idea in Figure 2 (namely, one should eliminate the components in X that are predictive of A only) and the proposed method, which does not specifically make an effort to eliminate these components. Specifically, the authors proposed to use a three-head dragonnet where two heads predict Q0 and Q1 and one head predicts A. In the end, only the estimated Q0 and Q1 are used for making the causal effect estimation. Should I understand that the estimated A in the dragonnet represents the components in X predictive of A? If that's the case, how can we ensure that the two heads for Q0 and Q1 do not use information in X_A? If not, how should we make the connection between the conceptual idea in Figure 2 and the implementation in Figure 3?*
> - The propensity score prediction head is just to prevent (implicit) regularization of the model from throwing away $X_{A\wedge Z}$ information that is necessary for identification. This is essentially a detail, and not key to the development of the paper. We have added a sentence in Section 4.2 "Q-Net" paragraph.
> 2. *related to (-1) above, it seems that a critical point is that Equation (3.3) Q(\tilde A, X) := E(Y | \tilde A,X) is in fact E (Y | \tilde A, X_A∧Z, X_Z). This seems to suggest that although conceptually CDE as in (3.2) should be based on X_A∧Z, X_Z which are unknown, in reality, one only needs to build models for Q(A,X) using the entire X. In this case, how does the proposed method differ from the naive method that "naively adjust for all the text as the confounding part"? Some clarification is needed.*
> - You're (mostly) right that the strategy for estimating $\eta(X)$ is just to build a model for $Q$ using the whole text $X$ (caveat below). The two observations here are: 1. this procedure actually does correspond to identifying a causal estimand (which is non-trivial! It's actually a CDE variant, not generally a simple ATE/ATT). 2. by viewing the modeling as a supervised representation learning step, we can use non-parametric propensity score estimation and double machine learning results to substantially robustify the estimate. There's also a nuance that the procedure for estimating $Q$ must preferentially use information in $A$ over information in $X_A$---see the discussion following Equation 4.3.
> 3. *It would be helpful to add the "naively adjust for all the text" method to the simulation, to see how this method is biased and problematic.*
> - We do report the "naively adjust for the text" with the outcome-only estimator as a baseline. Doing a "naively adjust for the text" method that also includes estimated propensity scores does not work at all, because the propensity scores for the "naively adjust for all the text" will be 0/1 (or very close to 0/1 in practice). Then, the double machine learning estimator is meaningless/of extreme value.
> 4. *Clarification needed: following Theorem 1, the author said "We emphasize that, regardless of whether the overlap condition holds or not, the propensity score of η(X) in condition 2 is accessible and meaningful." There is no η(X) in condition 2, and since η(X) is the outcome model, it does not have a propensity score.*
> - Sorry for the unclarity. There should be no "in condition 2". Theorem 1 intends to demonstrate that the two-dimensional variable $\eta(X)$ is a good confounding part under some reasonable assumptions. The propensity score of $\eta(X)$ is just $P(A=1|\eta(X))$ and it’s related to the propensity score $P(A=1|X_{A\wedge Z},X_Z)$ in condition 2 (See Equation B.1 in Appendix).
> 5. *abbreviation: On page 7, it says "AIPTW is double robust". AIPTW was not previously defined. AIPTW = augmented inverse probability T??? (treatment?) weighting? If you mean the estimator in (4.8), it should be said so.*
> - Sorry for the unclarity, AIPTW stands for Augmented Inverse Probability of Treatment weighted Estimator, which always refers to a semi-parametrically efficient or double machine-learning estimator.  The first abbreviation in the paper has been changed to its original name. Besides, the remark 3 on page 7 has been adjusted.
> 6. *bias: about Table 1, you said "bias of the new method is significantly lower than the bias of the outcome-only estimator". Bias can be both positive and negative, and hence lower bias is not necessarily better. Do you in fact mean average estimation error or root mean square estimation error? In fact, it would be good for you to define the bias that you are reporting in the paper.*
> - Thanks for pointing this out. Here, bias means average of absolute bias. We’ve changed the bias to absolute bias in the paper.

---

> > ### Author Response · Authors · 2022-11-11
> > **Authors' response**
> >
> > 7. *simulation. It is counter-intuitive that low confounding actually leads to a low coverage rate of the confidence intervals. In randomized trials (zero confounding), shouldn't the confidence interval be valid? Some explanation/discussion is needed.*
> > - We believe that this is a quirk of the way we do the simulation. When the confounding is low, there is not actually any relationship between the text and outcome (i.e., the Q model is bad, because there’s actually very little signal). This means that the estimate is dominated by noise in the model fit. In real problems, this should be less of an issue—if it’s impossible to predict $Y$ from the text, then presumably we’re not worried about confounding. But, we like it as a somewhat adversarial case for our method, illustrating a possible failure mode.
> > 8. *Please explain the specific differences between the current work and Claudia Shi, David M. Blei, and Victor Veitch. Adapting neural networks for the estimation of treatment effects. In Advances in Neural Information Processing Systems, 2019.*
> > - Shi et al examines efficient estimation of average treatment effects in the case where standard identification conditions (including overlap) hold with the observed covariates $X$. The setting, estimand, identification argument, and estimation procedure are all different. It is true that the neural architecture is the same here.
> > - We are making use of the idea from Shi et al of forcing the model to include propensity-score relevant information in its intermediate representation layer by using the propensity score prediction. This is an inessential detail in our paper, not important for any part of the development (this is following Pryzant et al, where it's used to improve the performance of the naive baseline). On the other hand, using an architecture that uses separate heads for treated and untreated units is important for us as a way to push the model to preferentially using treatment information---see the discussion following Equation 4.3.
> > 9. *I am not sure how novel is the proof of Theorem 2. The stated results seem to inherit from double machine learning in Chernozhukov's work. The author should comment on how their proof differs from that of Chernozhukov. This is important as this part is claimed to be the second main contribution of the paper.*
> > - There are two main novelties in our paper. The first is the apparent overlap violation setting---the fact that outcome modeling at the 1/4 rate suffices is mainly interesting because direct propensity score estimation isn't possible. (If it were, the usual double ML results would be better). The second is the observation that using a low dimensional sufficient statistic, we can very easily achieve the required 1/4 rate for the propensity score rate for DML results.
> > - Note that a main difference from Chernozhukov’s work is the setting. Here, we don’t have access to a directly estimated propensity score. The idea of estimating the propensity score from $\eta$ is only necessary because of our setting. The novel technical complications in the proof mainly relate to making sure that errors in Q estimation don’t excessively propagate to errors in the final estimator.

---

> ### Author Response · Authors · 2022-12-06
> **Authors' response**
>
> Do you have further concerns we can clarify? If we have fully addressed your concerns, we hope that you'll consider updating your score.

---

### Official Review · Reviewer_HJbF · 2022-10-22

**Confidence:** 4
**Correctness:** 4
**Technical Novelty And Significance:** 3
**Empirical Novelty And Significance:** 3
**Recommendation:** 6

**Clarity, Quality, Novelty And Reproducibility:**

The paper is clear and easy to follow.

Most of the novelty is in Theorem 1 and the architecture of Q-Net. The paper meets the bars of novelty.

**Strength And Weaknesses:**

### Strengths
* The problem is this paper is quite practical and interesting.
* Theorem 1 seems to be clever. It finds a sufficient statistic that blocks the paths that lead to violation of positivity.
* The paper is clear and easy to follow.

### Weaknesses
* Theorem 2 is a rather simple application of the DML approach. The theoretical results are also straightforward corollaries based on Chernokhukov et al (2016).

### Request for clarifications:
I am confused about several parts of the paper and I would like the authors to respond before I make the final decision.

1. **The positivity issue**: Given positivity violation, the estimator of $Q(\neg A, X)$ should be quite unreliable, where I assumed that the natural pair is $(X, A)$ and $\neg A$ is the treatment that is not observed. There should be very few examples of $X$ and $\neg A$. This might lead to violations of the assumptions of Theorem 2. Can you comment on this?
2. **Beginning of Section 4.2**: Why is $\eta$ a good sufficient statistic for $\hat{g}$?
3. **End of Section 3**: The phrase "propensity score of $\eta(X)$" does not make sense.
4. **Start of Section 4.1**: The phrase "naive outcome _regression_" is the right choice of words.

**Summary Of The Paper:**

This paper proposes a way to overcome the violation of positivity in causal inference with text data. Following the problem setup of Pryzant et al. (2020), a piece of text X contains both the treatment (assumed binary) and confounders. The authors find a sufficient statistic $\eta$ that prevents the violation of positivity. Another contribution of this paper is to combine the approach with the double machine learning idea and obtain easy valid confidence intervals.

**Summary Of The Review:**

This paper proposes a way to overcome the violation of positivity in causal inference with text data. Following the problem setup of Pryzant et al. (2020), a piece of text X contains both the treatment (assumed binary) and confounders. The authors find a sufficient statistic $\eta$ that prevents the violation of positivity. Another contribution of this paper is to combine the approach with the double machine learning idea and obtain easy valid confidence intervals.

---

> ### Author Response · Authors · 2022-11-10
> **Authors' response**
>
> 1. *Theorem 2 is a rather simple application of the DML approach. The theoretical results are also straightforward corollaries based on Chernokhukov et al (2016).*
> - We agree that the result is technically straightforward---though we see that as a feature! There are two main novelties here. The first is the apparent overlap violation setting---the fact that outcome modeling at the $\frac{1}{4}$ rate suffices is mainly interesting because direct propensity score estimation isn't possible. (If it were, the usual double ML results would be better). The second is the observation that by using standard non-parametric results and the 2-dimensional $\eta(X)$, we can easily achieve the required $\frac{1}{4}$ rate for the propensity score rate for DML results.
> 2. *The positivity issue: Given positivity violation, the estimator of $Q(\neg A,X)$ should be quite unreliable, where I assumed that the natural pair is $(X,A)$ and $\neg A$ is the treatment that is not observed. There should be very few examples of $X$ and $\neg A$. This might lead to violations of the assumptions of Theorem 2. Can you comment on this?*
> - The key assumption is that we have non-trivial overlap on $(X_{A\wedge Z}, X_Z, \neg A)$ tuples (even though we lack examples on $(X_A, X_{A\wedge Z}, X_Z, \neg A))$. In general, there is ambiguity when estimating $Q$ between using $X_A$ and $A$ information. That's why we use an architecture that forces the model to preferentially make use of $A$ information. See paragraph after Equation (4.3).
> 3. *Beginning of Section 4.2: Why is $\eta$ a good sufficient statistic for $\hat{g}$?*
> - $\eta(X)$ need not be a sufficient statistic for the propensity score $P(A=1|X_{A\wedge Z},X_Z)$ in the sense that it need not be the case that $P(A=1|\eta(X)) = P(A=1|X_{A\wedge Z},X_Z)$. However, $\eta(X)$ does suffice for causal identification (because it suffices to compute the outcome model). The proof of Theorem 1 shows that the causal estimand CDE is the same when we adjust for $(X_{A\wedge Z},X_Z)$ as the confounding part and adjust for $\eta(X)$ as the confounding part. By adjusting for $\eta(X)$, the CDE is identifiable from observational data.
> 4. *End of Section 3: The phrase "propensity score of $\eta(X)$" does not make sense.*
> - Sorry for the unclarity. There should be no "in condition 2". (The propensity score $P(A=1|X_{A\wedge Z},X_Z)$ in condition 2 is not the propensity score of $\eta(X)$.) The propensity score of $\eta(X)$ is defined as $P(A=1|\eta(X)))$ and it’s related to $P(A=1|X_{A\wedge Z},X_Z)$ in condition 2 (See Equation B.1 in Appendix).
> 5. *Start of Section 4.1: The phrase "naive outcome regression" is the right choice of words.*
> - Here we want to emphasize that in this estimator, no propensity score information is included, only the conditional outcomes. So we think the word "outcome-only" is more straightforward.

---

> ### Author Response · Authors · 2022-12-06
> **Authors' response**
>
> Do you have further concerns we can clarify? If we have fully addressed your concerns, we hope that you'll consider updating your score.

---

### Official Review · Reviewer_SeHw · 2022-10-25

**Confidence:** 2
**Correctness:** 3
**Technical Novelty And Significance:** 3
**Empirical Novelty And Significance:** 3
**Recommendation:** 6

**Clarity, Quality, Novelty And Reproducibility:**

In general, the paper is clear and the authors attempt to explain most of the methodology. Some concerns about clarity are raised in the section above.

**Strength And Weaknesses:**

Strengths: The evaluation of the proposed method shows a clear advantage over standard previous methods for estimation.

Weaknesses and Comments:
- CDE formulation: The authors state the following while explaining the formulation of the CDE expression on page 4: "our formal causal effect aims at capturing the effect of A through only the first, direct, path". Obviously, identifying the total effect of A on Y is different than identifying the direct effect, or at least the variant suggested in Equation 3.1. I'm missing the reasoning for why this is the purpose of the computation. Further clarification from the authors would be appreciated.

- Generality of the causal DAG (Figure 1): The model assumes the absence of confounding between $X$ and $Y$ and between $A$ and $Y$. In general, it is not clear how valid this assumption is for estimating the causal effects of attributes of a text document even if it is justified for the suggested problem of sentiment effect on sales.

- Citations: Most of the introduction states technical claims without any citation to back it. For example, the second paragraph discusses the identification of causal effects and the required conditions but there are no citations to support that.
Another example is in the following paragraph which states that "it is often reasonable to assume that text data has information about all common causes..."

- typo: p.5, "this estimator is yields".

**Summary Of The Paper:**

The paper addresses the problem of estimating the causal effect of an attribute of a text on some outcome variable under a setting where overlap is violated, i.e., the treatment variable is fully determined by the text features. Under the assumption that the problem satisfies the constraints of a given causal model (Figure 1), the authors propose an identification formula for the target effect that adjusts for part of the text to block confounding while resolving the overlap violation. Then, they propose an estimation procedure for this formula using standard double machine learning. Empirical evaluation shows the advantage of the proposed technique over baseline work.

**Summary Of The Review:**

The score reflects the concerns raised regarding the assumptions in the causal model and the derivation of the CDE expression which are discussed in the weaknesses section. I would be happy to revisit the score based on further clarification from the authors.

---

> ### Author Response · Authors · 2022-11-10
> **Authors' response**
>
> 1. *CDE formulation: The authors state the following while explaining the formulation of the CDE expression on page 4: "our formal causal effect aims at capturing the effect of $A$ through only the first, direct, path". Obviously, identifying the total effect of $A$ on $Y$ is different than identifying the direct effect, or at least the variant suggested in Equation 3.1. I'm missing the reasoning for why this is the purpose of the computation. Further clarification from the authors would be appreciated.*
> - The main reason for preferring this direct effect to a total effect is pragmatic: we show that the direct effect is identifiable under reasonable conditions (and the same seems unlikely to be true for the total effect because in general the decomposition of $X$ in Figure 2 is unknown). The idea here is that this formalization has reasonable fidelity to the question "what is the effect of attribute $A$ on outcome $Y$", and can be identified. (The general pattern of identifying or choosing a reasonable formalization that can be identified also motivates, e.g., the LATE in instrumental variables)
> 2. *Generality of the causal DAG (Figure 1): The model assumes the absence of confounding between $X$ and $Y$ and between $A$ and $Y$. In general, it is not clear how valid this assumption is for estimating the causal effects of attributes of a text document even if it is justified for the suggested problem of sentiment effect on sales.*
> - We agree that the assumption here is not always satisfied. Indeed, this has to be the case--causal inference from observational data is not always possible, so the identification assumptions need to be non-trivial. However, we think that the assumption is often reasonable--e.g., as in the example in the paper.
> 3. *Citations: Most of the introduction states technical claims without any citation to back it. For example, the second paragraph discusses the identification of causal effects and the required conditions but there are no citations to support that. Another example is in the following paragraph which states that "it is often reasonable to assume that text data has information about all common causes..."*
> - For the first one, we added the citation *Murphy, K. P. (2023). Probabilistic Machine Learning: Advanced Topics. MIT press* (See Chapter 36). For the second one, the observation is just that since the text contains all the information about the treatment, it, in particular, should contain all information in common to both the treatment and the outcome.
> 4. *typo: p.5, "this estimator is yields".*
> - Thanks for pointing this out. It has been changed.

---

> ### Author Response · Authors · 2022-12-06
> **Authors' response**
>
> Do you have further concerns we can clarify? If we have fully addressed your concerns, we hope that you'll consider updating your score.

---

### Official Review · Reviewer_m2tm · 2022-10-25

**Confidence:** 4
**Clarity, Quality, Novelty And Reproducibility:** See the next section for more details.
**Correctness:** 4
**Technical Novelty And Significance:** 3
**Empirical Novelty And Significance:** 3
**Recommendation:** 6

**Strength And Weaknesses:**

Strengths

-	The paper is very well-written. The proposed method is motivated and explained in a very nice way.

-	The paper addresses an interesting and important question: causal estimation with overlap violations.

-	The proposed method is very elegant conceptually.

-	The proposed method appears to have good performance both theoretically and empirically.

Weaknesses

-	The identification result is not new: it is the same as using prognostic score as deconfounding score in D’Amour & Franks (2021). See the next section for more details.


**Summary Of The Paper:**

The paper studies the problem of estimating causal effect with overlap violations. The authors focus on a specific application---causal estimation for text data. The authors start with writing down the model in terms of a DAG and discuss identification results based on the DAG. The authors then proceed with discussing estimation strategies; they propose an outcome only estimator and a doubly robust TI-estimator. They further establish a theorem that allows inference on the proposed TI-estimator. Finally, the authors demonstrate the performance of the proposed estimator through simulations and real data analysis.

**Summary Of The Review:**

Overall, I find this paper a good contribution to the literature. Here’re some comments/questions I have.

-	Correct me if I’m wrong, but the identification result is not new: think about the line of research in prognostic score. It will be helpful if the authors can discuss the connection of the paper to the prognostic score literature. The general theme of the proposed method is the same as using prognostic score as deconfounding score in D’Amour & Franks (2021). The authors discuss briefly on page 3 the differences between this paper and D’Amour & Franks (2021). To me, the paper is still a good one with an interesting application---text data and a nice neural network model, but emphasizing more on the differences and new contributions can help readers understand the novelty of the paper better.

-	Assumptions in Theorem 2. Do the two rates of convergence have to be o(n^(1/4))? Is it possible for the theorem to hold true if one is smaller while the other is larger (in the sense that as long as the product is o(n^(1/2)))? Thanks!

-	It will be helpful to state the full name of “CDE”.

-	There are some typos. Here are a few examples:

o	Page 3 line 3. “We assume there” -> “We do not assume”

o	Page 7 line 1. “It remains to given” -> “It remains to give”

---

> ### Author Response · Authors · 2022-11-10
> **Authors' response**
>
> Thank you for your support, and for your helpful comments and suggestions.
> 1. *Correct me if I’m wrong, but the identification result is not new: think about the line of research in prognostic score. It will be helpful if the authors can discuss the connection of the paper to the prognostic score literature. The general theme of the proposed method is the same as using prognostic score as deconfounding score in D’Amour & Franks (2021). The authors discuss briefly on page 3 the differences between this paper and D’Amour & Franks (2021). To me, the paper is still a good one with an interesting application---text data and a nice neural network model, but emphasizing more on the differences and new contributions can help readers understand the novelty of the paper better.*
> - D'Amour & Franks (2021) consider a case where there is overlap even with all the observed features. So, standard identification arguments apply in their settings. The motivation is mainly statistical efficiency. When adjusting for all observed features $X$ as the confounding part, extreme propensity scores will show up and hence lead to inaccurate causal estimations. In the text setting, if all the text is included as the confounding part (or we say include all observed features), overlap will collapse. We are obliged to give an alternative causal estimand and the identification argument for this causal estimand to be identified. Additionally, the main statistical robustness result in our paper– it suffices to estimate the outcome function at the $\frac{1}{4}$ rate – is also new.
> - On the other hand, D'Amour & Franks consider a continuous family of sufficient statistics (as opposed to the just 1 we consider), studying which gives optimal adjustment in the standard ATE estimation case.
> - We have updated the related work paragraph to make this clearer.
> 2. *Assumptions in Theorem 2. Do the two rates of convergence have to be o(n^(1/4))? Is it possible for the theorem to hold true if one is smaller while the other is larger (in the sense that as long as the product is o(n^(1/2)))? Thanks!*
> - Since the propensity score is estimated from the estimated conditional outcomes, a faster rate in propensity score can’t make up for a bad rate in the outcome model/Q model. On the other hand, the key observation of theorem 2 is that, because $\eta=(Q_0, Q_1)$ is only 2-dimensional, the propensity score can easily be estimated at the rate $\frac{1}{4}$ rate. This then allows us to tolerate rates as slow as $\frac{1}{4}$ on the outcome model/Q model.
> - So, a faster rate on the propensity score model wouldn’t help. A faster rate on the Q model would allow a slower rate on the propensity score model as long as the product is $o(n^{1/2})$. However, the propensity model is in two-dimensional space and the required $\frac{1}{4}$ rate can be easily achieved — so this relaxation also wouldn’t really help.
> - All above is with respect to the asymptotics in the theorem. In practice, however, faster rates would probably help considerably due to the finite-sample effects of higher-order terms.
> 3. *It will be helpful to state the full name of "CDE".*
> - Thanks for pointing this out. CDE is the controlled direct effect. We’ve changed the paragraph name from CDE to Controlled Direct Effect (CDE) on page 4.
> 4. *Typo 1: Page 3 line 3. “We assume there” -> “We do not assume”*
> - This is not a typo. I think we do assume that there exists a two-dimensional summary $\eta$ that suffices to handle confounding.
> 5. *Typo 2: Page 7 line 1. “It remains to given” -> “It remains to give”*
> - Thanks for pointing this out. It has been changed.

---

> ### Author Response · Authors · 2022-12-06
> **Authors' response**
>
> Do you have further concerns we can clarify? If we have fully addressed your concerns, we hope that you'll consider updating your score.

---

### Author Response · Authors · 2022-11-18
**Authors' response**

We thank all reviewers for their insightful comments and constructive suggestions. All reviewers agree that the paper addresses an interesting and important problem, and that it is clearly written. Some reviewers were concerned about the relationship with previous work; particularly with respect to the main estimation theorem and the architecture. We have clarified this in our detailed response to the reviews and made minor clarifying updates to the text of the paper.

---

### Decision · Program_Chairs · 2023-01-20

**Decision:**

Accept: poster

**Justification For Why Not Higher Score:**

In an abstract sense, the paper in much debt to other contributions on the lack of overlap problem, a large literature which contributes to other solutions to these problem. The manuscript acknowledges that other papers work on different assumptions that may relax some of the conditions in their presentation.

**Justification For Why Not Lower Score:**

The paper appropriately uses existing results in the literature to directly tackle a problem of practical relevance and test it with some less usual real-world dataset (not always a given in a treatment effect estimation contribution). Presentation also tries to tackle hard questions about the operational meaning of text in a causal effect estimation problem.

**Metareview: Summary, Strengths And Weaknesses:**

Causal effect estimation is particularly hard when treatments and confounders are text, which are high-dimensional and may be necessary to be used in their raw form, as abstract concepts that may underly a treatment-outcome relation and its common causes will not be observed directly. This paper addresses this issue by framing it in the context of the challenges of adjusting for confounders where the overlap under different treatment levels.

Strengths: the paper appropriately uses existing results in the literature to directly tackle a problem of practical relevance and test it with some less usual real-world dataset (not always a given in a treatment effect estimation contribution). Presentation also tries to tackle hard questions about the operational meaning of text in a causal effect estimation problem.

Weaknesses: in an abstract sense, the paper in much debt to other contributions on the lack of overlap problem, a large literature which contributes to other solutions to these problem. The manuscript acknowledges that other papers work on different assumptions that may relax some of the conditions in their presentation.

**Note From Pc:**

if the above contains the word "oral" or "spotlight" please see: "oral" presentation means -> notable-top-5% and "spotlight" means -> notable-top-25%. As stated in our emails, we are disassociating presentation type from AC recommendations